# Amplifying the effects of adding extra players during association football game-based scenarios

Diogo Coutinho[1,2,3]*, Bruno Gonçalves[4,5,6], Hugo Folgado[4,5], Bruno Travassos[2,6,7], Sara Santos[1,2], Jaime Sampaio[1,2]

1 Department of Sports Sciences, Exercise and Health, University of Trás-os-Montes and Alto Douro, Vila Real, Portugal, 2 Research Center in Sports Sciences, Health Sciences and Human Development, CIDESD, CreativeLab Research Community, Vila Real, Portugal, 3 University Institute of Maia, UMAIA, Maia, Portugal, 4 Departamento de Desporto e Saúde, Escola de Saúde e Desenvolvimento Humano, Universidade de Évora, Évora, Portugal, 5 Comprehensive Health Research Centre (CHRC), Universidade de Évora, Évora, Portugal, 6 Portugal Football School, Portuguese Football Federation, Oeiras, Portugal, 7 Department of Sports Sciences, University of Beira Interior, Covilhã, Portugal

* diogoamcoutinho@gmail.com

**Data Availability Statement:** In order to protect the subjects confidentiality and privacy, data are only available on request. Interested researchers may contact the board from the Research Center in

## Abstract

This study aimed to compare under-18 association football players' performance (age = 17.7±1.0 years; playing experience = 9.0 ± 3.2 years) when manipulating the number of teammates and opponents during football game-based practices. Time-motion, individual and tactical-related variables were monitored when manipulating conditions with different number of teammates and opponents (11vs11, No-Sup, No-Inf; 11vs12, Low-Sup, Low-Inf; 11vs13, Mod-Sup, Mod-Inf; and 11vs14, High-Sup, High-Inf). Results showed that adding teammates promoted increases in the longitudinal synchronization from No-Sup to Mod-Sup (Cohen's d with 95% of confidence intervals: 0.25 [0.12; 0.39]; p < .001) and High-Sup (0.61 [0.41; 0.82]; p < .001), while decreases in the distance to the nearest teammate, both in the offensive and defensive phases (p < .001 and p = .005, respectively). In addition, it was observed lower distance covered while running when playing in High-Sup compared to No-Sup (0.30 [-0.01; 0.61]; p = .002) during the defensive phase. Attacking in numerical infe-riority promoted a higher variability in the distance to the nearest teammate from No-Inf to High-Inf (0.83 [0.27; 1.38]; p = .044), while decreasing the physical demands, specifically distance covered while running (-0.49 [-0.99; 0.01]; p = .039). In turn, defending, mainly in high-inferiority, increased the total distance covered compared to No-Inf (0.61 [0.30; 0.91]; p < .001) and led to a decrease in the distance to the nearest teammate (-0.90 [-1.35; -0.44]; p = .002). Overall, coaches may manipulate the number of teammates and opponents to pro-mote distinct effects at the level of cooperation and opposition dynamical interactions.

## Introduction

In the past, association football training approaches were focused on the development of play-ers' physical, technical, perceptual and tactical skills using more repetitive training approaches

Sports Sciences, Health Sciences and Human Development to request access to the data (cidesd.geral@utad.pt).

**Funding:** This study was funded by CIDESD, supported by national funds through the Portuguese Foundation for Science and Technology, I.P., through a grant awarded to JS (UID04045/2020).

**Competing interests:** The authors report no conflict of interest.

and mostly based on exercises performed without opposition [1, 2]. For example, it was common to see training tasks in which 10 players develops a specific movement pattern without opposition, or with passive opposition, to facilitate a specific set of play or movement. However, despite the apparent success in the use of such process, the lack of informational references to guide individual and collective players' tactical behaviour, promote rigid set play or movements that cannot correspond to the level of variability and uncertainty of the game. As said, during competitive performances, based on spatial-temporal relations, players from one team promote cooperative interactions with teammates to achieve a common goal, such as progressing on the pitch and creating goal-scoring opportunities, while establishing competitive interactions with opponents to avoid their progression [1]. This highlights that players' actions on the pitch are dependent upon their ability to capture the relevant information from the environment [2–5]. Based on these premises, sports sciences investigation has been exploring different training approaches that may allow the players to develop their tactical behaviour [3].

Over the last years, there has been an increased focus on the use of game-based approaches, such as small-sided games (SSG), as it allows to concurrently develop the players' physical, technical and tactical skills while also coupling the players' actions to the environmental information [3, 6–8]. In addition, these game-based situations promote more dynamic and variable scenarios that allow developing players adaptability according to the changes in the information [3, 9, 10]. In fact, one reason for the increased popularity of this training approach is the possibility of manipulating its rules to emphasize specific information that will guide the players to the emergence of goal-directed behaviours [2, 4, 5]. For example, one rule manipulation that has been capturing attention from a wide body of research is the effects of playing in numerical superiority / inferiority during SSG [7, 9, 11–13]. Superiority during competitive performances may result from different aspects, such as a) numerical superiority as the team in possession overload a specific zone of the pitch (e.g., fullback and the winger against a defensive fullback); b) qualitative superiority, that refers to the ability of a team to emphasize the individual qualities of their players (e.g., creating space for 1vs1 of a highly skilled winger against a defender); or c) positional superiority, which means that the players positioning and body positioning would allow proper offensive passing lines (i.e., vertical and horizontal lines) [14]. While most of these superiorities may result from the players' individual quality and the team game model/playing system [1, 15], other types of superiorities may emerge such as a team playing with 10 players following a red card. Based on the previous insights, previous research explored how players' behaviour is modified by different numerical relations during SSG to aid coaches with practical and relevant information for practice design [6, 7, 9, 12, 15]. For instance, the available studies showed how different numerical relations, such as low inferiority/superiority scenarios (4vs3 and 5vs4) or high inferiority/superiority scenarios (5vs3 and 7vs4) influence players' positional and physical demands. Results revealed that playing in high superiority seems to promote team dispersion, decrease the distance to the opponent's goal and present a more regular positioning [6, 15] while playing under inferiority seems to promote team retraction and compactness [7, 13, 16]. In addition, SSG conditions with a low numerical difference between teams (e.g., 4vs3 and 5vs4) seem to be more physically demanding in contrast to SSG with higher numerical differences between teams (e.g., 5vs3 and 7vs4), as a result of the team under inferiority being able to perform additional effort to compensate the absence of one player [12]. These results highlight that different movement behaviours emerge because of different numerical unbalances, revealing a key constraint to be manipulated by the coaches. In fact, exposing players to higher numerical unbalances (i.e., difference between teams of more than one player, such as 5vs3, 7vs4) is a common approach used by coaches to develop the players' offensive and defensive behaviours in sectorial tasks [15].

Recently it was also observed that the use of numerical superiority in the practice of SSGs in comparison with numerical equality promoted differences in the learning process. Tasks with numerical superiority in attack should be used to develop the technical component, increasing players' participation in ball possession, while tasks in numerical equality may be considered to promote players' decision-making and adaptation to performance environments [17].

While research exploring the effects of different rules manipulation on SSG has been well established, little is known on how players adjust their movement behaviour during Large-Sided Games (LSG), where the players are more likely to act within similar playing roles as those found during the competition [11]. Apart from performing in their playing positions, LSG are also more suitable to simulate the technical (e.g., LSG allows a higher variability in the passing actions, such as long-distance and penetrative passes, while SSG emphasizes more short-distance passes) [18] and physical demands (i.e., acceleration distance, sprinting distance) of the competitive match play [19]. Coaches have been using LSG to shape the team tactical behaviour, whereas, playing under superiority or inferiority is often a rule adopted to decrease or increase the task perceptual-motor demands as well as to pursue offensive or defensive goals. For example, a recent study explored the players' positional behaviour during a 11vs10, aiming to simulate a possible early dismissal of a player from one team, with two additional conditions: i) defend close to their target or ii) high pressing [16]. In brief, it was found a decrease in the distance between teams, more regular movement behaviours and lower physical demands when pressing high compared to defending closer to the target [16]. Overall, these studies revealed important insights regarding players' movement behaviour adjustment during LSGs performed under low superiority / interiority scenarios. However, further research is required to understand the effects of playing in superiority / inferiority during LSGs, mainly using scenarios with different numerical unbalances (difference of 1, 2 or 3 players). However, most of the available scientific information had only addressed the difference of one player between teams when considering the development of offensive [11] or defensive behaviours [16]. While anecdotally, some reports have mentioned the use of additional players (i.e., higher than the 11vs11) [19], coaches may design tasks with extra players in a specific team (cooperation-perspective, 11+X vs 11) to emphasize possible local relations, while adding opponents (opposition-perspective, 11vs11+X) to amplify the perceptual demands and decrease the available space and time for the team under inferiority. Thus, it is required a better understanding on how players adjust their behaviour as a result of different manipulations, enhancing coaches' ability to tailor the game rules for specific aims. Therefore, this study aimed to compare players' physical, individual and collective tactical performance when manipulating the number of teammates and teammates during association football LSGs. It is hypothesized that major differences in players performance would emerge under high unbalance scenarios (i.e., High-Sup and High-Inf). In this sense, we expect to observe under High-Sup scenario a decrease in the distance between players and in the external load, while increasing the movement synchronization. In turn, it is hypothesised higher variability in the distance between players while attacking when facing a High-Inf scenario, whereas lower distance between teammates and physical demands while defending.

## Methods

### Participants

A total of twenty youth association football players from the same team (age = 17.7 ± 1.0 years; height = 176.4 ± 6.3 cm; weight = 63.8 ± 6.0 kg; playing experience = 9.0 ± 3.2 years) participated in this study. All players were engaged in four training sessions per week (90 to 105 minutes per session) and had an official 11-a-side match during the weekend at a regional playing

standard level. Two goalkeepers were part of the study but were excluded from the data analysis since their positioning is very restricted to a specific pitch area and their game dynamics are different from the outfield players. In addition, while three additional players were used (i.e., to promote the numerical unbalance between teams), their data was not computed as they did not participate in all conditions (e.g., the 11vs11 did not include any additional player). A written and informed consent was provided to the coaches, players, and by their legal guardians, as well as by the club, before the beginning of the study. All participants were notified that they could withdraw from the study at any time. The study protocol followed the guidelines and was approved by the local Ethics Committee of the Research Center in Sports Sciences, Health Sciences and Human Development (UIDB/4045/2020) and conformed to the recommendations of the Declaration of Helsinki.

## Study design

The head coach divided the players into 2 balanced teams of 10 players according to their playing positional role, as well as their physical, technical, and tactical performances. In addition, 3 players were assigned to participate in the game-based scenarios considering the number of additional players (see Fig 1), in which one was a central defender, the second was a midfielder and the last one a forward. The balanced game scenario (11vs11) was used as a control variable to understand the players' performance according to the increase in the number of teammates (cooperation-based perspective) and to the increase in the number of opponents (opposition-based perspective). Therefore, a total of 4 LSGs was performed twice in a randomised sequence, to ensure that both teams were exposed to an increase in the number of teammates and opponents (e.g., on the first day the team A played with the additional players, while in the second day it was the team B). For the cooperation-based perspective, the cooperation constraint effect was inspected by comparing the same ten players performance when playing without any additional teammate (no superiority, No-Sup), with 1 additional teammate (low superiority, Low-Sup), with 2 additional teammates (moderate superiority, Mod-Sup) and with 3 additional teammates (high superiority, High-Sup). In contrast, for the opposition-based perspective, the opponents' constraint effect was inspected when comparing the same 10

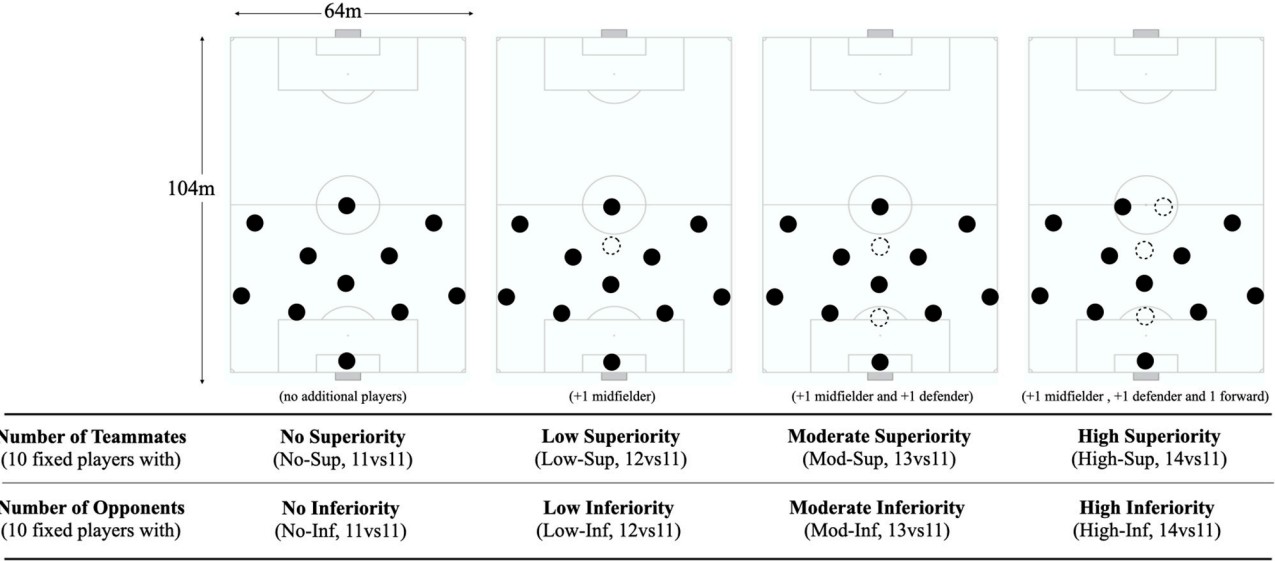

| Number of Teammates (10 fixed players with) | No Superiority (No-Sup, 11vs11) | Low Superiority (Low-Sup, 12vs11) | Moderate Superiority (Mod-Sup, 13vs11) | High Superiority (High-Sup, 14vs11) |
|---|---|---|---|---|
| Number of Opponents (10 fixed players with) | No Inferiority (No-Inf, 11vs11) | Low Inferiority (Low-Inf, 12vs11) | Moderate Inferiority (Mod-Inf, 13vs11) | High Inferiority (High-Inf, 14vs11) |

**Fig 1. Representation of LSG scenarios.**

players when confronted against 11 opponents (no inferiority, No-Inf), against 12 opponents (low inferiority, Low-Inf), against 13 opponents (moderate inferiority, Mod-Inf) and against 14 opponents (high inferiority, High-Inf).

## Procedures

All conditions were tested for two weeks. The first week was used for familiarization purposes, while the following week was used for the testing sessions. The experimental sessions were performed in non-consecutive days (i.e., with difference of two days between them) and developed during the middle of the in-season competitive period (season 2017/2018). The first session was used to familiarize the players with the different game-based scenarios. Both teams performed 2 halves of 20-min, where in each half one team played in numerical superiority / inferiority, that varied at each 5-min period. Then, the remaining two sessions were used to collect data. All sessions started with a 15-min warm-up based on low-intensity running, dynamic stretching and ball possession tasks. All game-based formats were performed on a 104x64m artificial turf pitch and were performed according to considering the official association football rules. Each game-based condition lasted for 10-min and was interspersed with a 5-min passive rest. During all conditions, each team was instructed by the coach that they should attempt to win the match, independently of numerical relation (e.g., inferiority) to avoid possible behaviours that lead the teams to preserve the result. Several association footballs were placed around the pitch to ensure its replacement as fast as possible, decreasing the time the ball was out of play. No coach feedback or encouragement was allowed during the conditions to avoid possible effects on the players' behaviours. Players were encouraged to hydrate by drinking water before the game-based scenarios and also in-between the bouts. Both sessions started at the same time of the day (18:00 hours) to avoid the effects of circadian rhythms and were completed within the same duration (~75minutes each session). Considering that the testing conditions were collected over two non-consecutive days in the same week, it allowed to expose the players to similar weather conditions (atmospheric temperature $14 \pm 3°$ C; humidity from 48% to 59%).

## Data collection and processing

Positional data and the distance covered during LSGs were gathered using 5 Hz Global Positioning System (GPS) units (SPI-PRO, GPSports, Canberra, ACT, Australia). The players' latitude and longitude information obtained with the GPS units were resampled to remove possible data gaps and to synchronize all the individual data. Following this procedure, the data were converted to meters using the Universal Transverse Mercator (UTM) coordinate system and a rotational matrix was applied to adjust the players displacement data, pitch length and width with the appropriate x and y-axis. This procedure was carried out by the data retrieved from 4 GPS units placed on each pitch corner [20]. In addition, all game-based scenarios were recorded using a digital video camera, Sony NV-GS230. The digital video camera was fixed at a 2-m height and aligned in the midfield part of the pitch. The video files were downloaded to a computer and notational analysis software (Longomatch, version 1.3.7., Fluendo) being later used to register the time of every ball-related action during the game-based conditions, following existing data processing procedures [21]. The following actions were considered: player gaining ball possession, player losing ball possession; player touching the ball without gaining possession; ball over the side line; ball over the end line; ball hitting the crossbar/post; ball shooting; goal scoring, and fouls. The position of the ball in the 2D horizontal plane was modelled according to an algorithm that integrated the player's relative positioning collected by the GPS system and the notational data resulting from the video analysis,

synchronizing the events with the GPS time. In addition, to recreate other non-GPS ball-specific positions, such as the goal, end and side lines, a total of 18 fixed locations were defined considering the pitch referentials.

All videos were analysed by an experienced performance analyst, and the data reliability was inspected by retesting 20% of the sample. The intraclass correlation was deemed as high (>0.93) [22].

## Physical variables

The total distance covered (per minute), the distance covered at different movement speed categories (per minute) for each player were also calculated [11]. The following speed categories were considered for analysis: walking (0.0–3.5 km/h); jogging (3.6–14.3 km/h), running (14.4–19.8 km/h), and sprinting (> 19.9 km/h) [11]. Based on the ball tracking, both the positional and physical-related variables were analysed according to the offensive and defensive phases.

## Individual tactical variables

For the individual tactical indicators, it was considered: (a) the number of completed passes; (b) number of forward, lateral and backward completed passes; (c) distance to the nearest opponent when passing for the total, forward, lateral and backward passes; (d) the number of dribbles; (e) total distance covered with the ball while dribbling; (f) average distance covered with the ball while dribbling; and (g) distance to the nearest opponent when dribbling [21]. For the passing direction classification, it was considered the following angles in relation to centre of the pitch and second quadrant: forward passes between 0° and 45°; lateral passes between 45° and 135°; and backward passes between 135° and 180° [21].

## Collective tactical variables

The positional data of the players were used to calculate intra-team coordination tendencies based on the time that players' dyads spent synchronized in both longitudinal and lateral directions. These variables were calculated with relative phase and the Hilbert transform [23]. The movement synchronization of each dyad was quantified by the percentage of time spent between -30° to 30° bin (near-in-phase mode of coordination) [20]. Also, data were used to assess the distance from each player to: (i) the nearest teammate and (ii) the nearest opponent, expressed by the absolute values (m) and the coefficient of variation (CV).

## Statistical analysis

The data were presented as means (M) ± standard deviations (SD). All data were assessed for outliers and assumptions of normality. Due to the existence of normal and non-normal distribution of data, the differences between conditions were assessed using parametric and non-parametric tests (ANOVA and Kruskal-Wallis, respectively) for each game scenario. Statistical significance was set at p < .05 and calculations were carried out using SPSS software V24.0 (IBM SPSS Statistics for Windows, Armonk, NY: IBM Corp.).

Complementary, pairwise differences (cooperation-perspective, No-Sup vs Low-Sup, No-Sup vs Mod-Sup, and No-Sup vs High-Sup; opposition-perspective, No-Inf vs Low-Inf, No-Inf vs Mod-Inf, and No-Inf vs High-Inf) were assessed via differences in group means expressed in raw data units with 90% confidence limits (CL). Thresholds for effect size statistics were: <0.2, trivial; <0.6, small; <1.20, moderate; <2.0, large; and >2.0, very large [24].

## Results

### Effects of playing in superiority during the offensive phase (cooperative-perspective)

The effects of the physical, individual and collective tactical variables when increasing the number of teammates during the offensive phase are presented in Table 1 and Fig 2. Interestingly, the physical-related variables revealed only decreases in the sprinting distance (Cohen's d with 95% of confidence intervals: 0.15 [-0.19; 0.50]; $X^2 = 8.94$, P = .039) from No-sup to Mod-Sup.

From the collective tactical perspective, it was found a increase in the longitudinal synchronization from No-Sup to Mod Sup (0.25 [0.12; 0.39]; $X^2 = 22.4$, P < .001) and to High-Sup (0.61 [0.41; 0.82]; $X^2 = 22.4$, P < .001). A decrease in the distance to the nearest teammate (-0.64 [-0.98; -0.31]; $X^2 = 22.0$, P < .001) was observed with an increase in the corresponding

**Table 1. Descriptive and statistical analysis for physical, individual and collective tactical-related variables when playing in superiority (cooperative-perspective) during the offensive phase.**

| Variables | Game-Based Conditions | | | | Difference in means (±90% CL) | | | P |
|---|---|---|---|---|---|---|---|---|
| | No-Sup (M ±SD) | Low-Sup (M ±SD) | Mod-Sup (M ±SD) | High-Sup (M ±SD) | No-Sup vs Low-Sup | No-Sup vs Mod-Sup | No-Sup vs High-Sup | |
| **Offensive Physical Variables** | | | | | | | | |
| Total Distance Covered (m) | 131.17±17.34 | 128.38±14.01 | 128.95±13.73 | 133.79±19.09 | -2.79; ±4.24 | -2.22; ±2.94 | 2.62; ±4.78 | .113 |
| Dist. Covered while Walking (m) | 6.26±2.95 | 6.11±3.02 | 6.39±2.62 | 6.74±3.33 | -0.15; ±0.6 | 0.13; ±0.52 | 0.48; ±0.77 | .400 |
| Dist. Covered while Jogging (m) | 94.70±14.03 | 95.98±14.19 | 96.75±15.88 | 92.77±15.44 | 1.28; ±2.61 | 2.04; ±3.55 | -1.93; ±3.44 | .259 |
| Dist. Covered while Running (m) | 19.79±9.23 | 18.46±8.96 | 19.35±8.43 | 22.71±10.22 | -1.33; ±3.49 | -0.43; ±2.77 | 2.93; ±2.96 | .121 |
| Dist. Covered while Sprinting (m) | 10.36±8.89 | 7.75±6.28 | 6.47±6.15 | 11.55±8.47 | -2.61; ±2.71 | -3.89; ±2.39 | 1.19; ±2.73 | **.039** |
| **Offensive Individual Tactical-Related Variables** | | | | | | | | |
| Total N° of Forward Passes (n) | 1.40±1.57 | 1.45±0.94 | 1.35±1.35 | 1.85±1.87 | 0.05; ±0.44 | -0.05; ±0.58 | 0.45; ±0.68 | .406 |
| Total N° of Lateral Passes (n) | 2.55±1.64 | 2.01±1.62 | 2.40±1.39 | 2.20±2.02 | -0.55; ±0.75 | -0.15; ±0.84 | -0.35; ±0.73 | .592 |
| Total N° of Backward Passes (n) | 1.20±1.01 | 1.50±1.19 | 1.25±1.07 | 1.20±1.24 | 0.30; ±0.68 | 0.05; ±0.48 | 0.01; ±0.61 | .882 |
| Dist. Nearest Opp. Frontal Passes (m) | 4.63±2.99 | 4.20±2.78 | 3.94±2.27 | 4.13±2.30 | -0.27; ±1.53 | -0.36; ±1.81 | -0.75; ±1.49 | .717 |
| Dist. Nearest Opp. Lateral Passes (m) | 4.12±1.95 | 4.64±2.50 | 3.58±1.57 | 5.80±3.29 | 0.31; ±1.07 | -0.42; ±1.12 | 1.46; ±1.79 | .591 |
| Dist. Nearest Opp. Backward Passes (m) | 2.55±1.36 | 3.66±3.35 | 3.20±2.50 | 4.89±4.54 | 1.13; ±2.12 | 0.84; ±1.58 | 2.77; ±3.54 | .074 |
| Total N° of Dribbles (n) | 5.60±3.10 | 7.75±4.68 | 6.30±2.85 | 7.75±6.58 | 2.15; ±2.29 | 0.70; ±1.62 | 2.15; ±2.03 | .761 |
| Average Dist. Covered in Dribble (m) | 7.43±5.41 | 6.23±3.16 | 5.52±2.80 | 5.89±3.91 | -1.21; ±2.19 | -2.07; ±1.64 | -1.60; ±2.46 | .274 |
| Dist. Nearest Opp. when dribbling (m) | 6.31±3.58 | 5.50±2.88 | 5.67±2.25 | 6.35±3.01 | -0.81; ±1.19 | -0.63; ±1.23 | 0.02; ±0.94 | .438 |
| **Offensive Collective Tactical-Related Variables** | | | | | | | | |
| Longitudinal Synchronization (%) | 72.11±8.05 | 74.15±10.94 | 74.91±9.90 | 78.87±14.01 | 2.04; ±1.73 | 2.8; ±1.49 | 6.75; ±2.24 | **< .001** |
| Lateral Synchronization (%) | 42.85±13.31 | 44.28±13.20 | 46.28±13.95 | 41.61±15.60 | 1.43; ±2.22 | 3.44; ±2.30 | -1.24; ±2.88 | .152 |
| Dist. to Nearest Teammate (m) | 10.39±1.55 | 9.90±1.87 | 9.13±2.20 | 8.91±1.86 | -0.49; ±0.58 | -1.27; ±0.65 | -1.49; ±0.46 | **< .001** |
| Dist. to Nearest Teammate (CV) | 39.47±4.67 | 40.25±7.57 | 41.05±6.60 | 45.50±8.42 | 0.78; ±2.98 | 1.58; ±2.89 | 6.03; ±3.24 | **.034** |
| Dist. to Nearest Opponent (m) | 6.61±2.21 | 6.68±2.17 | 6.60±1.99 | 7.16±2.99 | 0.07; ±0.43 | -0.02; ±0.47 | 0.55; ±0.60 | .923 |
| Dist. to Nearest Opponent (CV) | 53.81±5.51 | 56.36±9.90 | 58.23±12.82 | 54.99±6.08 | 2.55; ±3.31 | 4.42; ±4.50 | 1.17; ±2.01 | .461 |

**Note:** Dist, Distance; N°, Number; Opp, Opponent; CV, Coefficient of variation; CL, Confidence limits; Mod = Moderate; Sup = Superiority.

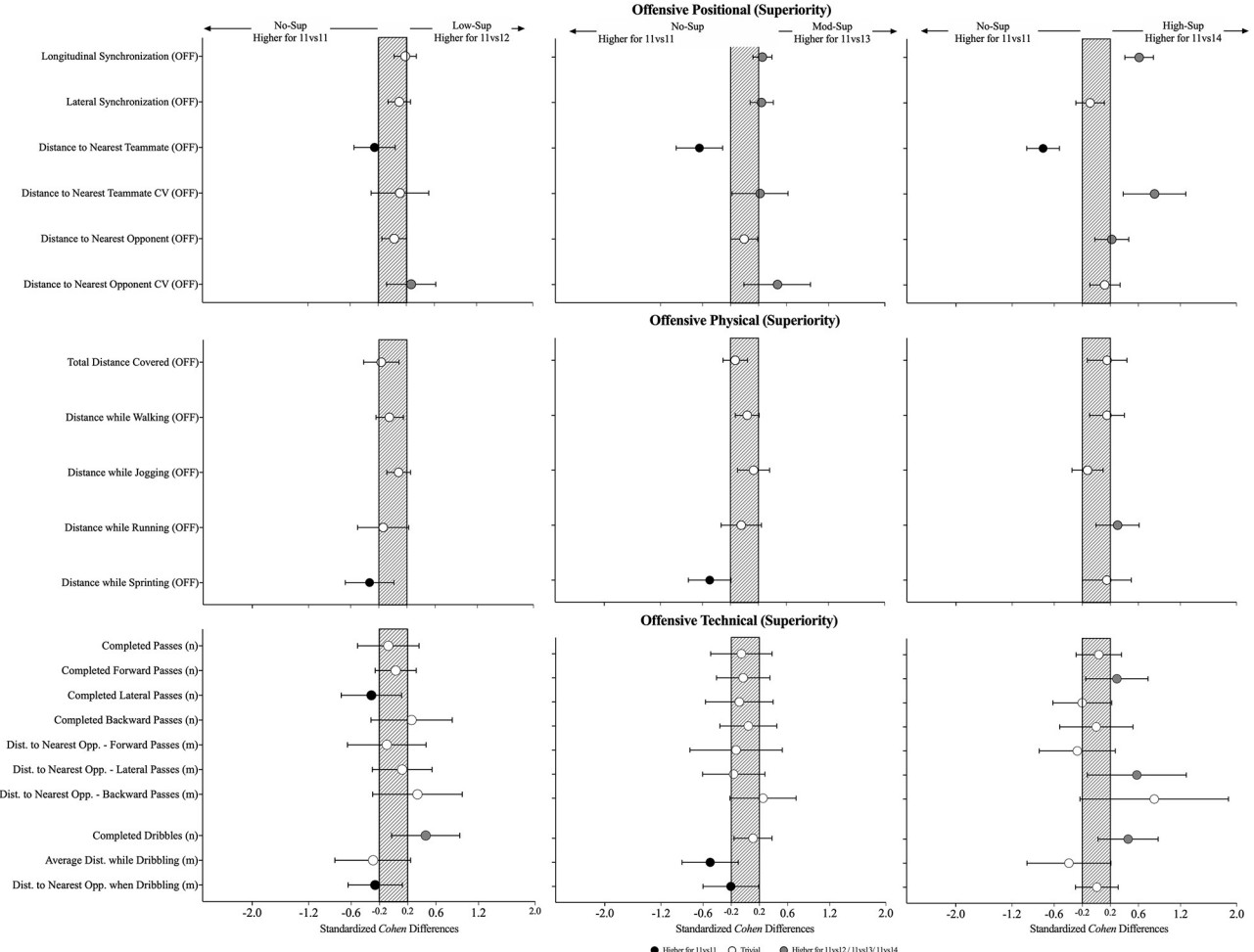

**Fig 2. Standardised (Cohen) differences in physical, individual and collective tactical variables considering the increase in the number of teammates (cooperation-perspective) during the offensive phase.** Error bars indicate uncertainty in the true mean changes with 90% confidence intervals. OFF = offensive.

CV (0.22 [-0.18; 0.62]; $X^2$ = 8.7, P < .034) from the No-Sup to Mod-Sup and an increase (CV) to High-Sup (0.83 [0.39; 1.28]; $X^2$ = 8.7, P < .034).

## Effects of playing in superiority during the defensive phase (cooperative-perspective)

The effects of the physical and individual and tactical variables when increasing the number of teammates during the defensive phase are presented in Table 2 and Fig 3. It was found an increase in the distance covered while walking from the No-Sup to the High-Sup (0.72 [0.30; 1.14]; F = 5.46.79, P = .008), in turn, there was a decrease in total distance covered (-0.79 [-1.19; -0.39]; F = 7.27, P = .002) and a decrease in the distance covered while running (-0.57 [-0.88; -0.26]; $X^2$ = 14.5, P = .002). Additionally, it was also observed a decrease in the distance covered while sprinting (1.77 [0.99; 2.55]; $X^2$ = 21.6, P < .001) in the No-Sup compared to the High-Sup. From the collective tactical performance while defending, the results showed increases in the longitudinal synchronization (0.43 [0.22; 0.65]; $X^2$ = 36.2, P < .001), a decrease in the lateral synchronization (-0.44 [-0.59; -0.28]; $X^2$ = 68.5, P < .001) from No-Sup to Low-

**Table 2. Descriptive and statistical analysis for physical and collective tactical-related variables when playing in superiority (cooperative-perspective) during the defensive phase.**

| Variables | Game-Based Conditions | | | | Difference in means (±90% CL) | | | P |
|---|---|---|---|---|---|---|---|---|
| | No-Sup (M ±SD) | Low-Sup (M ±SD) | Mod-Sup (M ±SD) | High-Sup (M ±SD) | No-Sup vs Low-Sup | No-Sup vs Mod-Sup | No-Sup vs High-Sup | |
| **Defensive Physical Variables** | | | | | | | | |
| Total Distance Covered (m) | 139.05±19.56 | 125.07±19.97 | 123.02±18.22 | 123.64±17.03 | -13.98; ±8.41 | -16.03; ±4.80 | -15.41; ±7.87 | **.002** |
| Dist. Covered while Walking (m) | 5.62±2.77 | 7.01±2.83 | 8.10±3.44 | 7.90±3.08 | 1.39; ±1.23 | 2.48; ±0.74 | 2.28; ±1.33 | **.008** |
| Dist. Covered while Jogging (m) | 95.89±16.49 | 94.01±18.14 | 88.63±16.92 | 87.54±16.09 | -1.88; ±8.27 | -7.26; ±3.54 | -8.34; ±6.64 | .126 |
| Dist. Covered while Running (m) | 26.40±11.72 | 17.98±9.45 | 19.58±11.68 | 20.18±8.71 | -8.42; ±3.23 | -6.82; ±3.53 | -6.22; ±3.43 | **.002** |
| Dist. Covered while Sprinting (m) | 11.15±7.67 | 6.05±4.91 | 6.71±7.23 | 71.35±64.29 | -5.10; ±2.67 | -4.44; ±3.15 | 60.21; ±26.5 | $< .001$ |
| **Defensive Collective Tactical-Related Variables** | | | | | | | | |
| Longitudinal Synchronization (%) | 70.12±10.62 | 75.57±16.33 | 75.71±8.58 | 79.62±13.27 | 5.45; ±2.72 | 5.59; ±1.91 | 9.49; ±2.23 | $< .001$ |
| Lateral Synchronization (%) | 60.42±15.79 | 54.01±14.69 | 58.91±14.51 | 45.09±13.30 | -6.41; ±2.26 | -1.51; ±2.76 | -15.33; ±2.97 | $< .001$ |
| Dist. to Nearest Teammate (m) | 9.11±2.04 | 8.21±1.29 | 7.92±1.45 | 7.86±1.26 | -0.90; ±0.58 | -1.18; ±0.77 | -1.25; ±0.56 | **.005** |
| Dist. to Nearest Teammate (CV) | 43.31±6.08 | 39.43±9.61 | 41.45±6.59 | 47.56±8.90 | -3.88; ±3.44 | -1.86; ±2.87 | 4.25; ±3.64 | $< .001$ |
| Dist. to Nearest Opponent (m) | 5.97±1.27 | 6.44±1.59 | 6.06±1.37 | 6.59±1.88 | 0.47; ±0.37 | 0.09; ±0.40 | 0.62; ±0.54 | .349 |
| Dist. to Nearest Opponent (CV) | 56.08±6.67 | 51.96±6.69 | 58.56±11.37 | 57.14±9.44 | -4.12; ±3.02 | 2.48; ±4.06 | 1.06; ±3.93 | **.040** |

**Note:** Dist, Distance; CV, Coefficient of variation; CL, Confidence limits; Mod = Moderate; Sup = Superiority.

Sup and a decrease from No-Sup to High-Sup (0.75 [0.57; 0.93]; $X^2 = 68.5$, P $< .001$). As well, it was found a decrease in the distance to the nearest teammate from the No-Sup to the High-Sup (-0.78 [-1.12; -0.43]; F = 7.79, P = .005). Finally, it was observed a decrease in the corresponding CV (-0.23 [-0.57; 0.12]; F = 6.65, P $< .001$) from the No-Sup to the Mod-Sup, while in turn there was an increase in the High-Sup (0.51 [0.07; 0.95]; F = 6.65, P $< .001$).

## Effects of playing in inferiority during the offensive phase (opposition-perspective)

The effects of the physical, individual and collective tactical variables when increasing the number of opponents are presented in Table 3 and Fig 4. In relation to the physical variables, the results showed high values in the No-Inf compared to the remaining conditions, mainly in the distance covered while running in which there was a decrease when comparing the No-Inf to the Mod-Inf (-0.62 [-0.95; -0.30]; F = 3.23, P = .039) and High-Inf (-0.57 [-0.88; -0.26]). The individual tactical variables revealed an increase in the number of completed backwards passes (0.25 [0.32; 0.82]; $X^2 = 8.62$, P = .035) from the No-Inf to the Low-Inf. From the collective tactical variables, it was identified a decrease in the longitudinal synchronization (-0.40 [-0.67; -0.12]; $X^2 = 18.0$, P $< .001$) when comparing the No-Inf with the Low-Inf, while in contrast there was an increase when comparing the No-Inf to the High-Inf. In addition, it was also identified a decrease in lateral synchronization (-0.43 [-0.62; -0.24]; $X^2 = 14.1$, P = .003) when comparing the No-Inf with the Low-Inf. The results also showed an increase in the coefficient of variation to the nearest teammate (0.83 [0.27; 1.38]; $X^2 = 8.1$, P = .044) from the No-Inf to

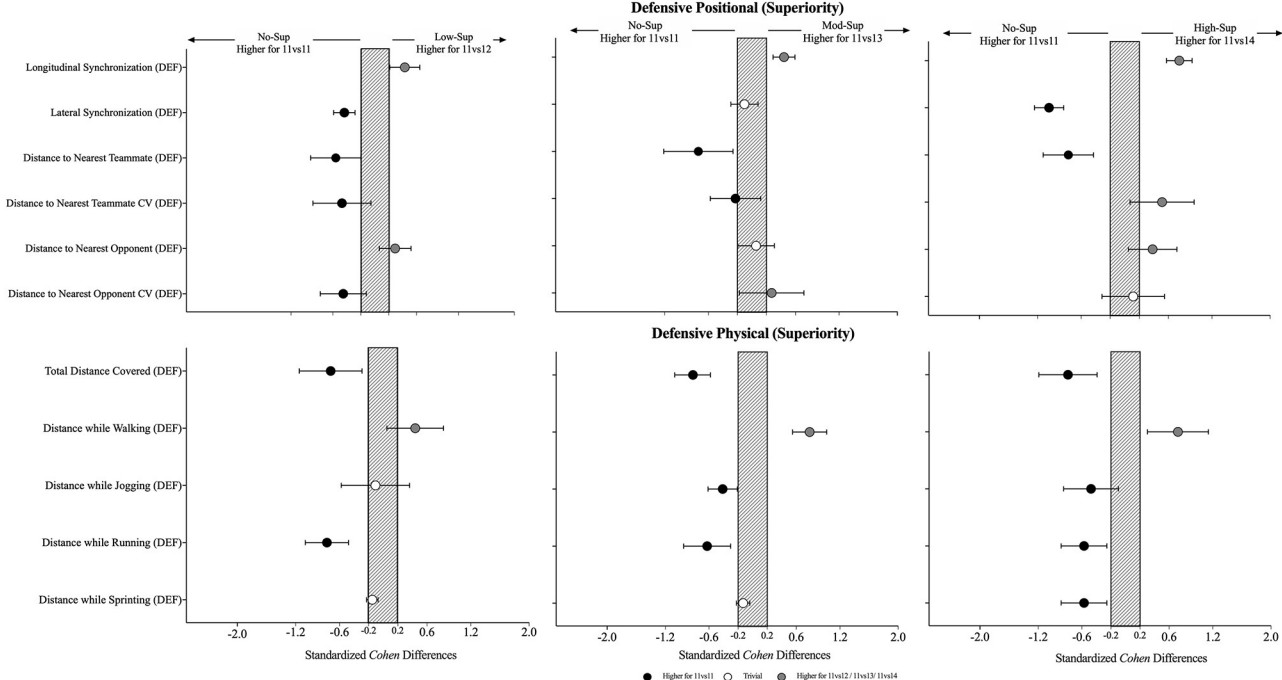

**Fig 3. Standardised (Cohen) differences in physical and collective tactical variables considering the increase in the number of teammates (cooperation-perspective) during the defensive phase.** Error bars indicate uncertainty in the true mean changes with 90% confidence intervals. OFF = offensive; DEF = defensive.

the High-Inf. In contrast, it was found an increase in the coefficient of variation for the distance to the nearest opponent (0.30 [-0.23; 0.65]; $X^2$ = 11.6, P = .009) for the same game-based scenarios.

## Effects of playing in inferiority during the defensive phase (opposition-perspective)

The effects of the physical and collective tactical variables when increasing the number of opponents are presented in Table 4 and Fig 5. From the physical perspective, the results showed an increase in the total distance covered (0.61 [0.30; 0.91]; F = 8.25, P < .001) and an increase in the distance covered while sprinting (0.49 [0.11; 0.87]; $X^2$ = 12.3, P = .006) when comparing the No-Inf to the High-inf. The results from the collective tactical data while defending showed a decrease in the longitudinal synchronization (-0.44 [-0.66; -0.21]; $X^2$ = 35.1, P < .001) when comparing the No-Inf to the Mod-Inf, however it was also identified an increase when contrasting the No-Inf to the High-Inf (0.44 [0.17; 0.70]). Also, it was found a decrease in the lateral synchronization (0.23 [0.05; 0.41]; F = 2.88, P = .046) from the No-Inf to the Low-Inf. The absolute distance to the nearest teammate ($X^2$ = 9.42, P = .002) revealed a trend to decrease when from the No-Inf to the High-Inf, mainly in the Mod-Inf (-0.27 [-0.48; -0.05]) and High-Inf (-0.30 [-0.54; -0.06]). Similarly, the absolute distance to the nearest opponent ($X^2$ = 16.6, P < .001) has also shown a decrease from the No-Inf compared to Low-Inf (-0.42 [-0.79; -0.06]), to Mod-Inf (-0.87 [-1.29; -0.45]) and to the High-Inf (-0.90 [-1.35; -0.45]).

**Table 3. Descriptive and statistical analysis for physical, individual and collective tactical-related variables when playing in inferiority (opposition-perspective) during the offensive phase.**

| Variables | Game-Based Conditions | | | | Difference in means (±90% CL) | | | P |
|---|---|---|---|---|---|---|---|---|
| | No-Inf (M ±SD) | Low-Inf (M ±SD) | Mod- Inf (M ±SD) | High- Inf (M ±SD) | No- Inf vs Low-Inf | No-Sup vs Mod-Inf | No-Inf vs High-Inf | |
| **Offensive Physical Variables** | | | | | | | | |
| Total Distance Covered (m) | 132.82±19.32 | 119.96±18.04 | 126.05±16.14 | 125.66±13.95 | -12.86; ±7.15 | -6.76; ±5.17 | -7.16; ±6.97 | **.034** |
| Dist. Covered while Walking (m) | 6.59±3.16 | 8.72±3.98 | 6.86±2.51 | 7.23±3.18 | 2.13; ±1.39 | 0.27; ±0.75 | 0.64; ±0.99 | .192 |
| Dist. Covered while Jogging (m) | 90.8±15.55 | 88.07±21.48 | 92.55±13.68 | 91.93±14.99 | -2.73; ±7.21 | 1.76; ±3.28 | 1.13; ±5.55 | .533 |
| Dist. Covered while Running (m) | 24.69±13.52 | 17.34±7.51 | 19.22±11.39 | 19.50±6.75 | -7.35; ±3.55 | -5.47; ±4.38 | -5.19; ±5.33 | **.039** |
| Dist. Covered while Sprinting (m) | 10.74±9.27 | 5.83±5.89 | 7.37±7.01 | 6.81±5.95 | -4.90; ±2.74 | -3.36; ±2.38 | -3.93; ±3.47 | .129 |
| **Offensive Individual Tactical-Related Variables** | | | | | | | | |
| Total N° of Forward Passes (n) | 1.25±1.62 | 1.20±1.06 | 1.30±1.22 | 1.15±0.88 | -0.05; ±0.69 | 0.05; ±0.73 | -0.10; ±0.78 | .562 |
| Total N° of Lateral Passes (n) | 1.80±1.54 | 2.20±1.96 | 1.95±1.43 | 1.40±1.19 | 0.40; ±0.77 | 0.15; ±0.62 | -0.40; ±0.57 | .249 |
| Total N° of Backward Passes (n) | 0.85±1.18 | 1.55±1.15 | 0.80±1.15 | 0.70±0.92 | 0.70; ±0.64 | -0.05; ±0.70 | -0.15; ±0.49 | **.035** |
| Dist. Nearest Opp. Frontal Passes (m) | 3.41±1.15 | 4.29±3.02 | 3.66±2.58 | 2.55±1.55 | 0.84; ±2.28 | 0.92; ±1.69 | -1.04; ±1.32 | .369 |
| Dist. Nearest Opp. Lateral Passes (m) | 4.08±2.20 | 4.22±2.49 | 2.50±1.53 | 4.01±1.65 | 0.44; ±1.42 | -1.42; ±1.48 | -0.23; ±1.26 | .086 |
| Dist. Nearest Opp. Backward Passes (m) | 4.63±4.32 | 4.58±3.60 | 3.78±1.79 | 4.39±2.82 | -0.43; ±4.43 | -2.67; ±6.65 | -1.46; ±5.94 | .896 |
| Total N° of Dribbles (n) | 7.79±3.71 | 5.43±3.22 | 6.37±2.70 | 6.72±3.91 | -2.36; ±1.56 | -1.71; ±1.95 | -1.36; ±1.90 | .195 |
| Average Dist. Covered in Dribble (m) | 5.59±2.79 | 6.30±2.39 | 5.49±3.14 | 5.92±3.75 | 0.71; ±1.36 | 0.01; ±1.49 | 0.44; ±1.66 | .274 |
| Dist. Nearest Opp. when dribbling (m) | 4.45±2.98 | 6.95±6.12 | 4.90±2.27 | 4.75±3.29 | 2.50; ±2.37 | 0.45; ±1.11 | 0.30; ±1.26 | .438 |
| **Offensive Collective Tactical-Related Variables** | | | | | | | | |
| Longitudinal Synchronization (%) | 74.8±10.32 | 69.53±17.77 | 73.41±11.76 | 77.71±11.52 | -5.27; ±3.67 | -1.39; ±2.44 | 2.91; ±2.63 | < **.001** |
| Lateral Synchronization (%) | 48.32±15.97 | 41.47±15.47 | 45.51±16.93 | 47.46±14.32 | -6.85; ±2.98 | -2.81; ±2.68 | -0.86; ±3.04 | **.003** |
| Dist. to Nearest Teammate (m) | 10.29±1.82 | 10.30±2.13 | 10.01±2.07 | 9.81±2.23 | 0.01; ±0.60 | -0.28; ±0.44 | -0.48; ±0.75 | .396 |
| Dist. to Nearest Teammate (CV) | 40.63±6.05 | 45.02±8.53 | 43.79±7.89 | 48.42±12.50 | 4.39; ±4.20 | 3.16; ±2.67 | 7.79; ±5.26 | **.044** |
| Dist. to Nearest Opponent (m) | 6.57±1.87 | 6.75±2.59 | 5.92±1.86 | 5.71±1.56 | 0.18; ±0.59 | -0.65; ±0.4 | -0.86; ±0.63 | **.005** |
| Dist. to Nearest Opponent (CV) | 57.01±8.18 | 56.09±9.07 | 64.01±8.58 | 60.04±12.01 | -0.91; ±4.54 | 6.99; ±4.32 | 3.03; ±5.01 | **.009** |

**Note:** Dist, Distance; N°, Number; Opp, Opponent; CV, Coefficient of variation; CL, Confidence limits; Mod = Moderate; Inf = Inferiority.

## Discussion

This study aimed to compare players' physical, individual and collective tactical performance when manipulating the number of teammates and opponents during association football LSGs. In general, higher effects were found when playing with high superiority and high inferiority. Accordingly, playing in superiority during the offensive phase promoted an increase in longitudinal synchronization, in the variability to the nearest teammates and opponents, as well as in the number of completed dribbles. During the defensive phase, adding teammates increased the longitudinal synchronization, but decreased the lateral synchronization and the physical demands. From the opposition perspective, playing in inferiority during the offensive phase led to a decrease in the distance to the nearest opponent, in the physical demands and distance covered with the ball. However, there was an increase in the variability in the distance to the nearest teammate and in the number of completed lateral passes. In turn, playing in inferiority while defending led to a higher absolute distance to the nearest teammates and opponents while increasing the physical demands, mainly the sprinting distance.

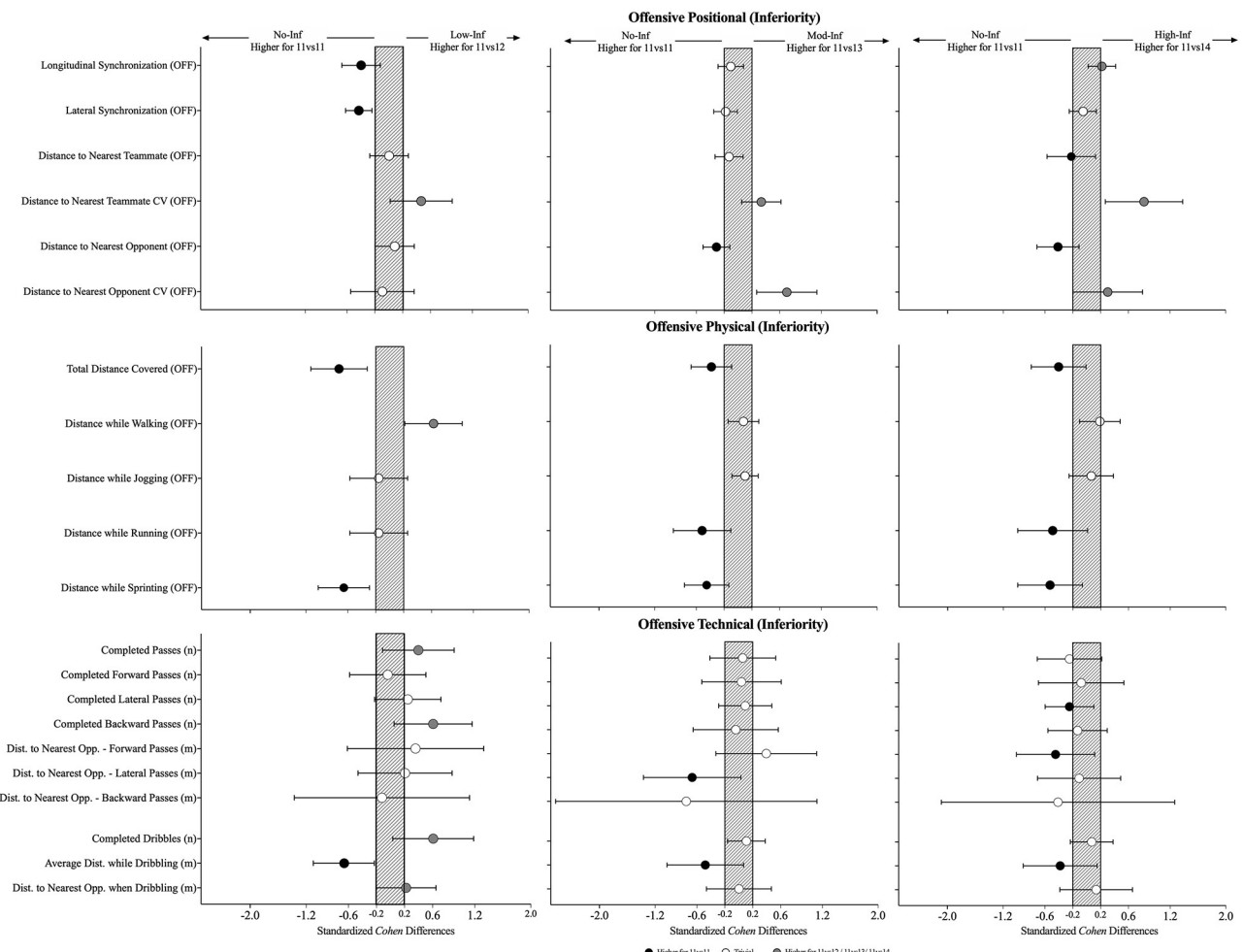

**Fig 4. Standardised (Cohen) differences in positional, physical and technical variables when playing in inferiority (opposition-perspective) during the offensive phase.** Error bars indicate uncertainty in the true mean changes with 90% confidence intervals. OFF = offensive; Dist = distance; Opp = opponent; n = number; m = meters.

## Effects of playing in superiority (cooperative-perspective, from 11vs11 to 14vs11)

From the cooperation perspective, playing in superiority contributed to an increase in the team longitudinal synchronization during the offensive phase, mainly during the Mod-Sup and High-Sup. This variable has been used to assess players' tactical performance, as players from the same team are likely to coordinate their movement behaviours to achieve a common specific goal [1, 20, 25]. The movement synchronization seems to be dependent on distance between players, as in general higher coordination has been found in closer distance between dyads [20, 23, 26]. Thus, the higher movement synchronization when additional teammates were included in the team (i.e., Mod-Sup and High-Sup), may contribute to decreasing the distance between players, favouring movement synchronization. In fact, the results from the distance to the nearest teammate support this evidence, as there was a trend to decrease this distance as more teammates were added during the game. Further, previous evidence also showed that playing in inferiority (i.e., 11vs10) during competitive performances leads the defensive team to adopt a low block and compact defensive strategy to face such difference

**Table 4. Descriptive and statistical analysis for physical and collective tactical-related variables when playing in inferiority (opposition-perspective) during the defensive phase.**

| Variables | Game-Based Conditions | | | | Difference in means (±90% CL) | | | P |
|---|---|---|---|---|---|---|---|---|
| | No-Inf (M ±SD) | Low-Inf (M ±SD) | Mod- Inf (M ±SD) | High- Inf (M ±SD) | No- Inf vs Low-Inf | No-Sup vs Mod-Inf | No-Inf vs High-Inf | |
| **Defensive Physical Variables** | | | | | | | | |
| Total Distance Covered (m) | 134.71±17.49 | 133.55±17.09 | 139.58±16.95 | 145.30±15.51 | -1.16; ±3.27 | 4.87; ±2.98 | 10.59; ±5.32 | < .001 |
| Dist. Covered while Walking (m) | 6.19±2.88 | 6.03±2.49 | 5.07±2.64 | 5.41±1.92 | -0.16; ±0.54 | -1.12; ±0.65 | -0.77; ±0.68 | **.008** |
| Dist. Covered while Jogging (m) | 94.52±14.94 | 97.51±15.05 | 95.93±19.01 | 99.31±17.61 | 2.99; ±2.42 | 1.41; ±4.22 | 4.79; ±4.32 | .440 |
| Dist. Covered while Running (m) | 25.03±10.81 | 23.91±9.91 | 25.79±9.79 | 27.91±9.27 | -1.12; ±3.68 | 0.76; ±3.66 | 2.87; ±3.48 | .371 |
| Dist. Covered while Sprinting (m) | 8.94±7.05 | 5.92±4.84 | 12.71±7.35 | 12.46±7.87 | -3.03; ±2.36 | 3.77; ±3.16 | 3.51; ±2.71 | **.006** |
| **Defensive Collective Tactical-Related Variables** | | | | | | | | |
| Longitudinal Synchronization (%) | 80.08±8.93 | 79.35±8.01 | 76.30±8.29 | 83.90±9.10 | -0.74; ±1.75 | -3.78; ±1.97 | 3.81; ±2.30 | < .001 |
| Lateral Synchronization (%) | 56.87±12.14 | 59.90±11.97 | 55.99±11.81 | 59.51±16.16 | 3.04; ±2.33 | -0.88; ±2.54 | 2.64; ±3.37 | **.046** |
| Dist. to Nearest Teammate (m) | 8.97±1.66 | 8.67±1.75 | 8.46±1.84 | 8.39±2.01 | -0.3; ±0.43 | -0.5; ±0.41 | -0.58; ±0.46 | **.024** |
| Dist. to Nearest Teammate (CV) | 41.19±7.08 | 42.62±5.63 | 43.76±5.13 | 42.89±7.24 | 1.43; ±2.83 | 2.57; ±2.62 | 1.70; ±3.46 | .088 |
| Dist. to Nearest Opponent (m) | 5.81±0.96 | 5.46±0.70 | 5.08±0.83 | 5.05±0.75 | -0.36; ±0.31 | -0.74; ±0.35 | -0.76; ±0.38 | < .001 |
| Dist. to Nearest Opponent (CV) | 53.42±7.91 | 54.45±7.76 | 54.57±8.89 | 55.18±7.15 | 1.03; ±3.24 | 1.15; ±3.16 | 1.76; ±3.71 | .854 |

**Note:** Dist, Distance; CV, Coefficient of variation; CL, Confidence limits; Mod = Moderate; Inf = Inferiority.

between teams. In turn, the offensive team needs to create variability in passing actions to move the defensive block and create space [27]. Similar findings were identified in the presented study, whereas there were completed more forward passes in the High-Sup.

During the attacking phase, differences in physical performance were mostly found in the distance covered while sprinting. For instance, sprinting has been identified as a key movement that leads to the creation of goal-scoring opportunities [28]. Consequently, players have to move to perceive and be able to maintain high levels of movement coordination [29]. Playing against a high block may afford then the players to perform sprinting actions in the back of the defensive line because of the distance between defenders to the goal, while the same pattern may be required against a deep-defending team, whose compactness may limit space between defensive lines (i.e., distance between the defenders and midfielders). So, the available space to perform and the defensive team strategy seems to impact the players' physical performance, positional demands and technical actions when attempting to create goal-scoring opportunities [30].

The effects of playing in superiority while defending revealed interesting results, as it was found an increase in the longitudinal synchronization when playing in numerical superiority while defending. Previous research showed that when the difference in the number of players between teams increases, the team in superiority tends to pressure and force the opposing team to retract close to their own goal, allowing also to control the game pace and reducing the physical demands [16]. The variations found in the team longitudinal synchronization while defending under different numerical relations seems to result from the defensive strategy, as

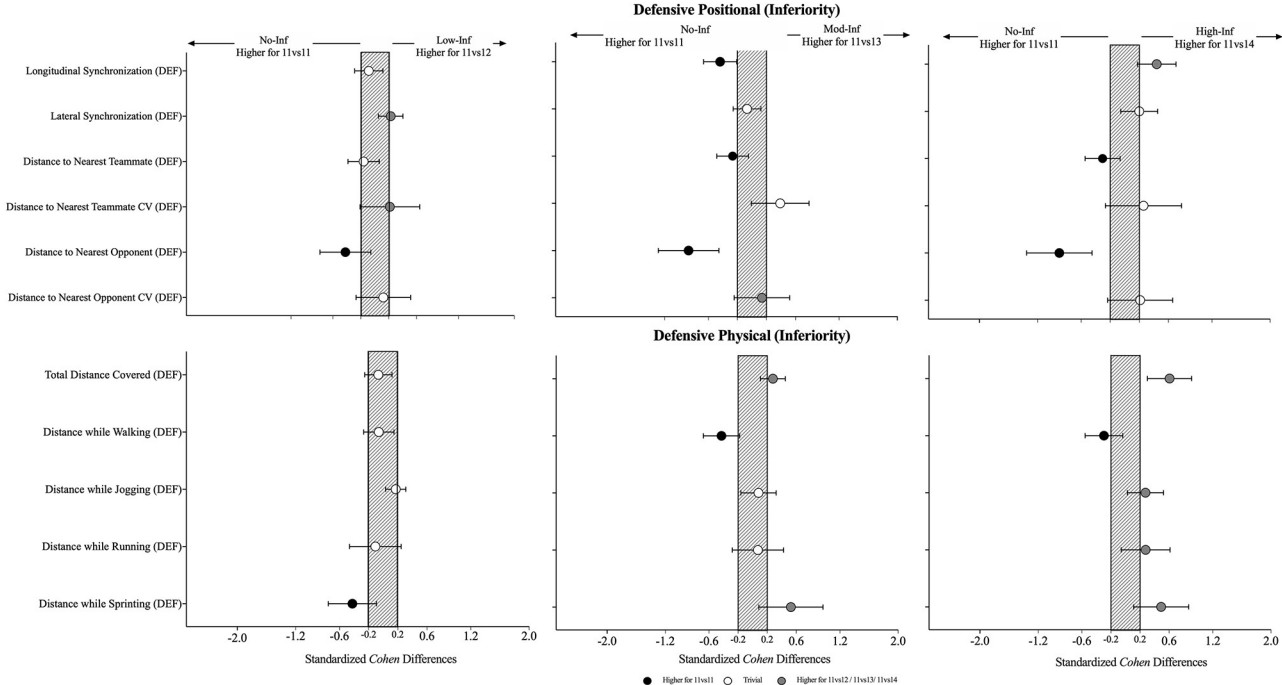

**Fig 5. Standardised (Cohen) differences in physical and collective tactical variables considering the increase in the number of opponents (opposition-perspective) during the defensive phase.** Error bars indicate uncertainty in the true mean changes with 90% confidence intervals. Dist = distance; Opp.

the team defensive length is one sensitive variable when varying from high-press to a more compact defensive strategy [30].

Unsurprisingly, there was a decrease in distance to the nearest teammate as the number of teammates increased. However, the analysis of its variability returned different trends that seem to reflect the players' adaptability to the task constraints manipulation [31]. That is, players seem to adopt different adaptive behaviours when playing under different numerical relations. For example, an increase pressuring from the opposing team is expected when facing extra players, while a more conservative approach may emerge under numerical balance between teams [27]. As a consequence, increasing this variability resulted as a functional adjustment according to local adaptations and players' density that allowed to maintain the team stability to the different configurations of play [9].

## Effects of playing in inferiority (opposition-perspective, from 11vs11 to 11vs14)

From the opposition perspective, playing in inferiority during the offensive phase revealed a general decrease in team longitudinal synchronization, mainly during the Low-Inf. A common strategy adopted by teams when playing in inferiority (i.e., difference of one player) while attacking is exploring counter-attacks or direct-plays, which contribute to a decrease in ball possession time [6, 13]. Consequently, it may limit the opportunities for the team to maintain movement synchronization in the longitudinal direction. Indeed, the lower values found for the physical variables during the offensive phase and the lower distance covered while dribbling during the inferiority situations (Low-Inf, Mod-Inf and High-Inf) in relation to the No-Inf, may suggest lower ball possession when facing numerical inferiority situations. In contrast, higher values in longitudinal synchronization when facing the High-Inf condition were found.

As players are likely to adopt more direct solutions of play, such as playing a long ball when playing in inferiority [27] and facing a high press [30], it may be possible that during conditions of high-inferiority this option emerges more often. This idea may be strengthened by the lower distance to the nearest opponent when performing frontal passes, and that may result in an attempt from the opposing team to pressure the player with the ball to avoid them performing these types of passes. On the other hand, it may also contribute to the increase in the variability in the distance to the nearest teammate, which may reflect an adaptive functional behaviour from the offensive team as a result of the increased pressure from the opponent [6, 30].

When additional opponents were added to the task, it was found a decrease in the distance to the nearest teammate and in the distance to the nearest opponent while defending. These evidence highlights a general trend to decrease team dispersion when playing in inferiority as an attempt to cover and deny space to opposing players [16, 27]. Players are more synchronized when playing closer [2, 20, 26], however, different results emerged in this study during the defensive phase. In general, the defensive players present high levels of movement coordination in the lateral coordination to limit the offensive team space in the wide channels [20, 32]. For instance, there were clearly lower values for lateral and longitudinal synchronization during the Mod-Inf scenario compared to the remaining conditions. In this condition, it was added an additional central defender to the opposing team, which may have limited effect on the defensive team under inferiority. That is, by being in inferiority, the defensive team is likely to retreat and protect the goal [27]. In turn, the additional centre back may only amplify back passing options, which by being further to the defensive team goal, may limit the impact on their defensive behaviour. In contrast, the additional midfielder and mostly the forward, may induce more offensive disruptive patterns amplifying defending movement synchronization.

From the physical perspective, the condition that elicited higher physical demands was the High-Inf, 11vs14, which contrasted with the results identified in previous research. For example, literature reported higher physical demands during conditions of low-inferiority on the basis that the team playing with fewer players performs an additional effort to compensate for playing with one fewer player [12]. This difference may be linked with increased movement synchronization, that is, players are required to move to maintain the levels of movement coordination, that were higher in the High-Inf scenario [2]. However, these differences may also be linked to the inclusion of one additional forward, which are usually players that are responsible to explore free space and unpredictable movements to break the alignment with the defenders [33], and may have afforded the team in possession to explore more offensive movements, consequently increasing the physical demands of the defensive.

While this study adds novel and important findings regarding the effects of including additional teammates and opponents, some limitations should be acknowledged. For example, previous studies showed that players of different expertise levels responded differently to the manipulation of the number of teammates and opponents during SSGs [6]. Future studies should then explore how players' performance from different expertise levels may be affected by the manipulation of the number (superiority vs inferiority) during LSGs. In addition, this study added a specific playing position for each condition (11vs12, midfielder; 11vs13, midfielder + central defender; 11vs14, midfielder + central defender + forward). Under this possibility, the results may be interpreted with caution as it is likely that some responses resulted from the inclusion of these playing positions, and so, future studies should also consider including players with more playing roles or performing as neutral players to provide support to each team while attacking. In this respect, it is also important to note that the scenarios explored in this work are less representative of the competitive demands. For instance, further research can explore how the confrontation between team sectors (i.e., defenders vs attackers,

### Summary of Main Findings from the Effects of Adding Teammates (Cooperation) and Opponents (Opposition)

**Fig 6. Summary of main findings from adding teammates (cooperation perspective) or opponents (opposition perspective) during game-based scenarios on the physical, individual and collective tactical behaviours.** Dist = distance; Avg = average; n = number; m = meters.

midfielders vs midfielders) or even the use of different numerical relations to support the emergence of goal-direction behaviours (e.g., attempting to build-up from the back against defenders under different numerical relations such as Gk+7vs6+Gk, Gk+7vs7+Gk, or Gk +7vs8+Gk). Despite these considerations, previous reports suggests that coaches may apply variability during the training sessions to enhance players' adaptability [34], and thus coaches may consider these manipulations to foster adaptive movement behaviours. Finally, and considering the role of the modern goalkeepers in association football, future research should consider their positioning, as it is likely that distinct coordination patterns would emerge as result of the different numerical relations between teams.

## Conclusions

Major effects were identified during conditions where there were higher numerical differences between teammates (high-superiority and high-inferiority, see Fig 6 for a summary of main effects).

Coaches can use low difference scenarios (Low-Sup and Low-Inf) to increase the perceptual demands while maintaining similar behaviours as those found in more balanced scenarios. In contrast, playing in high-superiority (High-Sup) may be beneficial during congested fixtures, in which one team may be composed by players that competed in the last match (11 line-up players plus 3 substitute players), as it found to decrease the physical demands and increase the team synchrony, which seems to be impaired during these tight calendar schedules [23]. As well, coaches may use high-inferiority scenarios (High-Inf) to prepare players to face opponents of higher quality, as it seems to promote more compact behaviours, amplify movement synchronization, and explore offensive adaptive behaviours.

## Author Contributions

**Conceptualization:** Diogo Coutinho, Bruno Gonçalves, Sara Santos.

**Data curation:** Diogo Coutinho, Hugo Folgado.

**Formal analysis:** Diogo Coutinho, Bruno Gonçalves, Bruno Travassos, Sara Santos, Jaime Sampaio.

**Funding acquisition:** Jaime Sampaio.

**Investigation:** Diogo Coutinho, Bruno Gonçalves, Bruno Travassos, Sara Santos.

**Methodology:** Diogo Coutinho, Bruno Gonçalves, Hugo Folgado, Bruno Travassos, Sara Santos.

**Project administration:** Bruno Travassos, Jaime Sampaio.

**Resources:** Hugo Folgado, Bruno Travassos, Sara Santos.

**Software:** Diogo Coutinho, Bruno Gonçalves, Hugo Folgado.

**Supervision:** Jaime Sampaio.

**Validation:** Diogo Coutinho, Bruno Gonçalves, Hugo Folgado, Sara Santos, Jaime Sampaio.

**Visualization:** Diogo Coutinho, Bruno Gonçalves, Hugo Folgado.

**Writing – original draft:** Diogo Coutinho, Bruno Gonçalves, Hugo Folgado, Sara Santos.

**Writing – review & editing:** Bruno Travassos, Jaime Sampaio.

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
