## [Decision Letter · Decision Letter 0]

21 Mar 2022

PONE-D-22-04617Amplifying the effects of playing with additional teammates (or opponents) during association football game-based scenariosPLOS ONE

Dear Dr. Coutinho,

Thank you for submitting your manuscript to PLOS ONE. After careful consideration, we feel that it has merit but does not fully meet PLOS ONE’s publication criteria as it currently stands. Therefore, we invite you to submit a revised version of the manuscript that addresses the points raised during the review process.

We look forward to receiving your revised manuscript.

Kind regards,

Rabiu Muazu Musa, PhD

Academic Editor

PLOS ONE

Journal Requirements:

[NO authors have competing interests]. 

Additional Editor Comments:

Please ensure that the manuscript is thoroughly proofread before resubmission

Reviewers' comments:

Reviewer's Responses to Questions

**Comments to the Author**

1. Is the manuscript technically sound, and do the data support the conclusions?

Reviewer #1: Yes

Reviewer #2: Yes

2. Has the statistical analysis been performed appropriately and rigorously? 

Reviewer #1: Yes

Reviewer #2: Yes

3. Have the authors made all data underlying the findings in their manuscript fully available?

Reviewer #1: No

Reviewer #2: Yes

4. Is the manuscript presented in an intelligible fashion and written in standard English?

Reviewer #1: No

Reviewer #2: Yes

5. Review Comments to the Author

Reviewer #1: Manuscript Number: PONE-D-22-04617

Manuscript Title: Amplifying the effects of playing with additional teammates (or opponents) during association football game-based scenarios

This is an interesting study with meaningful information to practical field of soccer training. The experimental design and methods adopted are technical sounds. I believe the outcomes of this study enriching our current understanding of game-based training/large size game of soccer training. However, some area needed to clarify and justify the practical implantation in term of technical and tactical coherence of soccer training.

1. My first comment would like to address the ecological models of 11 vs 11,12,13,14 as a game-based training. The team formation during offensive and defensive phases could be critical factors to create superiority and inferiority scenarios. As the study added a central defender, a central midfielder, and a striker. Ultimately, we can understand technical and tactical behaviors of the players. Such information needed to be clarified in the manuscript.

2. This study excluded the goalkeeper performance for data analysis. The coordination of defensive lines and team balance with GK is fundamental principle during inferior number of play or high pressure defending. This consideration should be highlighted in your study.

3. In the present study, the authors discussed the performance of small-sided games in comparison to game sciences. I think small-sided game training does not approach to the team performance as entire picture of soccer game. For example, the high speed running/sprints and technical performance are absolutely different to real game. The authors are encouraged to use large-size game training to develop the background information/discussion of the study.

4. Please concise the interpretations of results and discussion. Some sentences are redundant and are required grammatical check.

Reviewer #2: Manuscript Number: PONE-D-22-04617

Manuscript Title: Amplifying the effects of playing with additional teammates (or opponents) during association football game-based scenarios

I would like to congratulate the authors for the work done. If on a first instance I was septic about the work, after I carefully analysed it, I believe it is a good piece of information for coaches to use in the future. The paper is generally really well written and easy to follow, the methods well presented as also as the use of pictures and tables, and I would recommend publication after a minor revision.

Firstly, the authors should be consistent in the terminology used:

- Field or pitch

- Football or association football (or even perhaps soccer)

I would then start by the title. Isn’t the title too long? Can the author think of a shorter title, perhaps? Suggestion: “Amplifying the effects of extra players during association football game-bases scenarios”.

Abstract

Line 48: replace “youth players” by “under-18 players”.

Line 50: replace “both more and fewer” by “different”

Introduction

Line 81: remove “on the field”.

Line 86: I would remove “such as small-sided games (SSG)”, as it is not relevant to the main topic.

Line 95: it then makes sense to refer to SSGs. Make sure you use small-sided games terminology before SSG if you decided to remove the previous term.

Line 125 to 136: I would carefully consider what the authors mentioned. I am not sure if coaches use more than 11 players on their tasks, as it tends to lose task representativeness. Particular, in a youth level, where coaches struggle to have 22 players fit for sessions. I believe authors should rephrase this idea. This approach it is interesting in scenarios where coaches actually have more than 22 players so they can keep players engaged and active in training, leading then to the exploration of different behaviours according with the numerical unbalance. And this idea should also lead to the authors hypothesis (which they did not include). Having this previous idea, it would be interesting to include a couple of hypothesis after the objective.

Study design

Line 161: include large-sided games before LSGs.

Line 166: replace “with none” for “without”.

Procedures

Line 194: The author well controlled several variables. Considering all previous procedures, did the authors also were able to control weather conditions? Were the weather conditions similar for every day of data collection? And if not, did this have any effect on the results?

Discussion

Lines 428 – 434: Wasn’t the authors’ idea a result of the different playing density instead? More players on the pitch (more than in the real game scenario), led to different adaptations (pressing may be one of them).

Line 490: One of the biggest limitations to this study is this is not an ecology task, and therefore, must be carefully planned according with the coach/team environment. Why haven’t authors refer to this?

Line 504: As well, future research should be conducted to understand the tactical development of using game-based scenarios in between sectors, e.g., defenders vs attackers, midfielders vs midfielders. And the implication of using different numerical relations to the development of different playing styles, e.g., progressive progression, direct game, pressing, etc.

Conclusions

Generally, I believe the authors repeated themselves in this section. The results of this study are really novice and I would replace the whole section by recommendations for coaches. The authors should consider recommend certain numerical relationship to develop certain aspects of the game in the training session. For instance, by using 11vs14, I will expect my players to behave as … Also, when coaches look to develop pressing principles, they can make use of … The idea would be to provide coaching guidelines that could be related to figure 6 and summarise the main the practical main findings of this work.

Well done for the work completed!

Best wishes.

6. PLOS authors have the option to publish the peer review history of their article (what does this mean?). If published, this will include your full peer review and any attached files.

Reviewer #1: **Yes: **Yung-Sheng Chen

Reviewer #2: No

---

## [Author Response · Author response to Decision Letter 0]

6 Apr 2022

Dear Editor,

Thank you very much for your reassessment of our manuscript entitled “Amplifying the effects of extra players during association football game-bases scenarios”. We thank the Reviewers and to the Associate Editor for their positive and constructive feedback that have greatly contributed to the manuscript improvement. We appreciate the time and effort that you and the reviewers team dedicated to providing feedback valuable information to improve our work. We have included most of the suggestions made by the reviewers. Those changes are highlighted in blue color or track in change within the manuscript.

Reviewer: 1

Comments to the Author

This is an interesting study with meaningful information to practical field of soccer training. The experimental design and methods adopted are technical sounds. I believe the outcomes of this study enriching our current understanding of game-based training/large size game of soccer training. However, some area needed to clarify and justify the practical implantation in term of technical and tactical coherence of soccer training.

Authors: Dear reviewer. We would like to express our gratitude for such insightful and constructive suggestions that we believe that improved our work. We have carefully adjusted the manuscript according to the reviewer suggestions. 

Reviewer: 1

My first comment would like to address the ecological models of 11 vs 11,12,13,14 as a game-based training. The team formation during offensive and defensive phases could be critical factors to create superiority and inferiority scenarios. As the study added a central defender, a central midfielder, and a striker. Ultimately, we can understand technical and tactical behaviors of the players. Such information needed to be clarified in the manuscript.

Authors: Thank you for this insightful comment. We have attempted to address both in the introduction and the discussion sections, how different superiorities may be achieved and the impact of the playing role on the players’ performance. 

Reviewer: 1

This study excluded the goalkeeper performance for data analysis. The coordination of defensive lines and team balance with GK is fundamental principle during inferior number of play or high pressure defending. This consideration should be highlighted in your study.

Authors: Thank you for pointing this out. It was added as limitation and further study perspective. “Also, future research should consider the goalkeeper positioning, as it is likely that distinct coordination patterns would emerge as result of the different numerical relations between teams.” 

Reviewer: 1

In the present study, the authors discussed the performance of small-sided games in comparison to game sciences. I think small-sided game training does not approach to the team performance as entire picture of soccer game. For example, the high speed running/sprints and technical performance are absolutely different to real game. The authors are encouraged to use large-size game training to develop the background information/discussion of the study.

Authors: We agree with reviewer’s assessement. We have adjusted the information on the introduction to reinforce the role of LSG in preparing the players for competition. “Apart from performing in their playing positions, LSG are also more able to simulate the technical (e.g., LSG allows a higher variability in the passing actions, such as long-distance and penetrative passes, while SSG emphasizes more short-distance passes) (19) and physical demands (i.e., acceleration distance, sprinting distance) of the competitive match play (20). Coaches have been using LSG to shape the team tactical behaviour, whereas, playing under superiority or inferiority is often a rule adopted to decrease or increase the task perceptual-motor demands. Thus, a better understanding on how players adjust their behaviour as result of different manipulations would enhance the coach’s ability to tailor the game rules according to specific aims.”

Reviewer: 1

Please concise the interpretations of results and discussion. Some sentences are redundant and are required grammatical check.

Authors: We really appreciate the reviewer suggestions across the manuscript. We have carefully adjusted the discussion to better meet the results findings. We also have adjusted the English grammar across the manuscript. We sincerely appreciate all valuable comments and suggestions. 

Reviewer #2:

I would like to congratulate the authors for the work done. If on a first instance I was septic about the work, after I carefully analysed it, I believe it is a good piece of information for coaches to use in the future. The paper is generally really well written and easy to follow, the methods well presented as also as the use of pictures and tables, and I would recommend publication after a minor revision.

Authors: We highly appreciate the reviewer insightful and helpful comments on our manuscript, that we believed that help us to better tailor its information. 

Reviewer #2:

Firstly, the authors should be consistent in the terminology used:

- Field or pitch

- Football or association football (or even perhaps soccer);

Authors: Thank you for pointing this out. It was adopted the term pitch and association football. 

Reviewer #2:

I would then start by the title. Isn’t the title too long? Can the author think of a shorter title, perhaps? Suggestion: “Amplifying the effects of extra players during association football game-bases scenarios”. 

Authors: we agree with the reviewer, and we have adapted our title as suggested. 

Reviewer #2:

Abstract

Line 48: replace “youth players” by “under-18 players”.

Authors: We appreciate. It was modified accordingly. 

Line 50: replace “both more and fewer” by “different”

Authors: Thank you. Changed accordingly. 

Reviewer #2:

Introduction

Line 81: remove “on the field”.

Authors: Thank you. It was changed accordingly. 

Reviewer #2:

Line 86: I would remove “such as small-sided games (SSG)”, as it is not relevant to the main topic. 

Authors: We appreciate. According to the reviewer following suggestion, we decided to keep this section to better frame the information on line 95. 

Reviewer #2:

Line 95: it then makes sense to refer to SSGs. Make sure you use small-sided games terminology before SSG if you decided to remove the previous term.

Authors: As stated in the previous comment, we maintained the information to better frame our intention in this section. 

Reviewer #2:

Line 125 to 136: I would carefully consider what the authors mentioned. I am not sure if coaches use more than 11 players on their tasks, as it tends to lose task representativeness. Particular, in a youth level, where coaches struggle to have 22 players fit for sessions. I believe authors should rephrase this idea. This approach it is interesting in scenarios where coaches actually have more than 22 players so they can keep players engaged and active in training, leading then to the exploration of different behaviours according with the numerical unbalance. And this idea should also lead to the authors hypothesis (which they did not include). Having this previous idea, it would be interesting to include a couple of hypotheses after the objective.

Authors: We agree with the reviewers’ perspective. In this respect, we have modified this section for the following: “However, most of the available scientific information had only addressed the difference of one player between teams when considering the development of offensive (13) or defensive behaviours (18). While anecdotally, some reports have mentioned the use of additional players (i.e., higher than the 11vs11) (20), coaches may design tasks with extra players in a specific team (cooperation-perspective, 11+X vs 11) to emphasize possible local relations, while adding opponents (opposition-perspective, 11vs11+X) to amplify the perceptual demands and decrease the available space and time for the team under inferiority. However, , more research is required to better understand the real effects of adding teammates and opponents during large game-based situations.”

In addition, the following hypotheses were considered: 

“It is hypothesized that major differences in players performance would emerge under high unbalance scenarios (i.e., High-Sup and High-Inf). In this respect, we expect that under High-Sup scenario it is found a decrease in the distance between players and in the external load, while an increase the movement synchronization. In turn, it is hypothesised higher variability in the distance between players while attacking when facing a High-Inf situation, whereas lower distance between teammates and physical demands while defending.” 

Reviewer #2:

Study design

Line 161: include large-sided games before LSGs.

Authors: We appreciate the reviewer suggestion. It was added accordingly. 

Reviewer #2:

Line 166: replace “with none” for “without”.

Authors: Thank you for pointing this out. It was modified. 

Reviewer #2:

Procedures

Line 194: The author well controlled several variables. Considering all previous procedures, did the authors also were able to control weather conditions? Were the weather conditions similar for every day of data collection? And if not, did this have any effect on the results?

Authors: Thank you for raising up this question. Both testing sessions were performed in the same week, with one day apart, allowing to have similar weather conditions between the two testing days. The following information was added:

“Both sessions started at the same time of the day (18:00 hours) to avoid the effects of circadian rhythms and were completed within the same duration (~75minutes each session). Considering that the testing conditions were collected over two non-consecutive days in the same week, it allowed to expose the players to similar weather conditions (atmospheric temperature 14 ± 3º C).”

Reviewer #2:

Discussion

Lines 428 – 434: Wasn’t the authors’ idea a result of the different playing density instead? More players on the pitch (more than in the real game scenario), led to different adaptations (pressing may be one of them).

Authors: We appreciate this insightful comment. We have modified this section to guide it more towards players’ adaptability to players’ density as the follows: 

“That is, players seem to adopt different adaptive behaviours when playing under different numerical relations. For example, an increase pressure from the opposing team is expected when facing extra players, while a more conservative approach may emerge under numerical balance between teams (28). As a consequence, increasing this variability resulted as a functional adjustment according to local adaptations and players’ density that allowed to maintain the team stability to the different configurations of play (11).”

Reviewer #2:

Line 490: One of the biggest limitations to this study is this is not an ecology task, and therefore, must be carefully planned according with the coach/team environment. Why haven’t authors refer to this?

Authors: Thank you for this suggestion. We have highlighted it during the limitations section. The following sentence was added: “In this respect, it is also important to note that the scenarios explored in this work are less representative of the competitive demands, however, previous reports suggests that coaches may apply variability during the training sessions to enhance players’ adaptability (37), and thus coaches may consider these manipulations to foster adaptive movement behaviours.” 

Reviewer #2:

Line 504: As well, future research should be conducted to understand the tactical development of using game-based scenarios in between sectors, e.g., defenders vs attackers, midfielders vs midfielders. And the implication of using different numerical relations to the development of different playing styles, e.g., progressive progression, direct game, pressing, etc.

Authors: We do agree with the reviewer perspective, and thus the following sentence was included in the limitations / future research sections: “For instance, further research exploring how the confrontation between team sectors (i.e., defenders vs attackers, midfielders vs midfielders) or even the use of different numerical relations to support the emergence of goal-direction behaviours (e.g., attempting to build-up from the back against defenders under different numerical relations such as Gk+7vs6+Gk, Gk+7vs7+Gk, or Gk+7vs8+Gk).”

Reviewer #2:

Conclusions

Generally, I believe the authors repeated themselves in this section. The results of this study are really novice and I would replace the whole section by recommendations for coaches. The authors should consider recommend certain numerical relationship to develop certain aspects of the game in the training session. For instance, by using 11vs14, I will expect my players to behave as … Also, when coaches look to develop pressing principles, they can make use of … The idea would be to provide coaching guidelines that could be related to figure 6 and summarise the main the practical main findings of this work.

Authors: Thank you for pointing this out. We have modified the discussion section by attempting to provide a more practical perspective on the main findings. The following section was added: “Accordingly, coaches may use low difference scenarios (Low-Sup and Low-Inf) to increase the perceptual demands while maintaining similar behaviours as those found in more balanced scenarios. In contrast, playing in high-superiority (High-Sup) may be beneficial during congested fixtures, in which one team may be composed by the players that competed in the last match (11 line-up players plus 3 substitute players), as it found to decrease the physical demands and increase the team synchrony, which seems to be impaired during these tight calendar schedules (38). In turn, coaches may use high-inferiority situations (High-Inf) to prepare the players to face opponents of higher quality, as it seems to promote more compact behaviours, amplify movement synchronization, and also explore offensive adaptive behaviours.” 

Reviewer #2:

Well done for the work completed!

Best wishes.

Authors: We would like to thank the reviewer for the effort and expertise that contributed towards the improvement of our manuscript. Thank you.

---

## [Decision Letter · Decision Letter 1]

18 Apr 2022

PONE-D-22-04617R1Amplifying the effects of adding extra players during association football game-based scenariosPLOS ONE

Dear Dr. Coutinho,

Thank you for submitting your manuscript to PLOS ONE. After careful consideration, we feel that it has merit but does not fully meet PLOS ONE’s publication criteria as it currently stands. Therefore, we invite you to submit a revised version of the manuscript that addresses the points raised during the review process.

We look forward to receiving your revised manuscript.

Kind regards,

Rabiu Muazu Musa, PhD

Academic Editor

PLOS ONE

Journal Requirements:

Additional Editor Comments (if provided):

**Comments to the Author**

1. If the authors have adequately addressed your comments raised in a previous round of review and you feel that this manuscript is now acceptable for publication, you may indicate that here to bypass the “Comments to the Author” section, enter your conflict of interest statement in the “Confidential to Editor” section, and submit your "Accept" recommendation.

Reviewer #1: All comments have been addressed

Reviewer #2: All comments have been addressed

2. Is the manuscript technically sound, and do the data support the conclusions?

Reviewer #1: Yes

Reviewer #2: Yes

3. Has the statistical analysis been performed appropriately and rigorously? 

Reviewer #1: Yes

Reviewer #2: Yes

4. Have the authors made all data underlying the findings in their manuscript fully available?

Reviewer #1: Yes

Reviewer #2: Yes

5. Is the manuscript presented in an intelligible fashion and written in standard English?

Reviewer #1: No

Reviewer #2: Yes

6. Review Comments to the Author

Reviewer #1: I am pleased to see the authors improved the quality of the manuscript with practical aspects in relation to technical/tactical behavior of LSG soccer training. I only have several minor comments this time.

Abstract. Please add information of studying participants. Furthermore, add statistical report in the result section. Please also clarify “small to moderate effects”.

Add references to support the statement, line 109-112.

Line 122. what do you mean about “higher numerical unbalances”.

Line 175, you have 10 players for each team (please add information related to their training status). Should the authors include 3 additional players as studying participants?

Line 216, .what is the resting interval between the testing days?

Line 228, “officially sized” is redundant.

Line 237, add humidity of the testing days.

Lin2, 249, the location of filming? I suppose to be the central of the ground if only one camera.

Table 1, p value of Dis Covered while sprint should be 0.039.

Clarify small/moderate, small to moderate …effect.

X2 should be superscript. Revised in the result section.

Line 478, what do you mean “as the pitch was kept constant”?

Line 523, remove “available”.

Line 500, revise “balls” to “ball”.

Concise the sentences, line 513-516 and conclusion.

Reviewer #2: I would like to congratulate the authors with the final manuscript. I believe they addressed all my comments/suggestions and I think the final version is really good. Well done!

---

## [Author Response · Author response to Decision Letter 1]

23 May 2022

Dear Editor,

Thank you very much for your reassessment of our manuscript entitled “Amplifying the effects of extra players during association football game-bases scenarios”. We really would like to thank the Reviewers and to the Associate Editor for being always constructive in their feedback. We have included most of the suggestions made by the reviewer. Those changes are highlighted in blue color or track in change within the manuscript.

Journal Requirements:

Authors: Dear team, some references were included following the revision #1 to support requests from the reviewers, where we attempted to improve our theoretical rationale. In the current review, no additional references were added to the text. 

Comments to the Author

Reviewer #1: I am pleased to see the authors improved the quality of the manuscript with practical aspects in relation to technical/tactical behavior of LSG soccer training. I only have several minor comments this time.

Authors: We highly appreciate the reviewer insightful and helpful comments on our manuscript. We do believe that it helped us to improve the work, and consequently, improving its clarity and rationale. 

Reviewer #1

Abstract. Please add information of studying participants. Furthermore, add statistical report in the result section. Please also clarify “small to moderate effects”.

Authors: Thank you. We have added authors age and experience. Also, statistical report was included, while the small to moderate effects were modified by the corresponding values of Cohen’s d effects. Again, thank you for helping us in the manuscript improvement. 

Reviewer #1

Add references to support the statement, line 109-112.

Authors: We appreciate. References were accordingly added: “Based on the previous insights, previous research explored how players’ behaviour is modified by different numerical relations during SSG to aid coaches with practical and relevant information for practice design (6, 7, 9, 12, 15)”

Reviewer #1

Line 122. what do you mean about “higher numerical unbalances”.

Authors: Our intention it was to mention to differences higher than one player per team. The following information was added to the sentence: “In fact, exposing players to higher numerical unbalances (i.e., difference between teams of more than one player, such as 5vs3, 7vs4) is a common approach used by coaches to develop the players’ offensive and defensive behaviours in sectorial tasks (15).”

Reviewer #1

Line 175, you have 10 players for each team (please add information related to their training status). Should the authors include 3 additional players as studying participants?

Authors: We do understand the reviewer perspective. Despite being used 25 players (2 Goalkeepers, 20 outfield players and the 3 additional players), only the data from the 20 were used as consisted in the players presented in all data collection sessions and conditions. The following sentence was added: “In addition, while three additional players were used (i.e., to promote the numerical unbalance between teams), their data was not computed as they did not participate in all conditions (e.g., the 11vs11 did not include any additional player).”

Also, the following information was added to the participants training status: “All players were engaged in four training sessions per week (90 to 105 minutes per session) and had an official 11-a-side match during the weekend at a regional playing standard level.”. 

Furthermore, it was added during the procedures the phase level of the competition: “All conditions were tested for two weeks. The first week was used for familiarization purposes, while the following week was used for the testing sessions. The experimental sessions were performed in non-consecutive days (i.e., with difference of two days between them) and developed during the middle of the in-season competitive period (season 2017/2018).

Reviewer #1

Line 216, .what is the resting interval between the testing days?

Authors: Thank you for raising this question. It were collected with a difference of 2 days. 

Reviewer #1

Line 228, “officially sized” is redundant.

Authors: Thank you. It was deleted accordingly. 

Reviewer #1

Line 237, add humidity of the testing days.

Authors: We appreciate this suggestion. It was accordingly added. 

Reviewer #1

Lin2, 249, the location of filming? I suppose to be the central of the ground if only one camera.

Authors: Yes. It was used only one, and in this respect, it was placed in the central zone of the pitch at a height of 2m. The following information was added: “The digital video camera was fixed at a 2-m height and aligned in the midfield part of the pitch.”

Reviewer #1

Table 1, p value of Dis Covered while sprint should be 0.039.

Authors: Thank you for identifying this mistake. We have updated the table accordingly. 

Reviewer #1

Clarify small/moderate, small to moderate …effect.

Authors: We do appreciate the reviewer corrections. We have modified it accordingly. To avoid possible mistakes, we have updated as in the abstract. That is, the information was modified by the corresponding Cohen’s d values. 

Reviewer #1

X2 should be superscript. Revised in the result section.

Authors: Thank you. It was modified accordingly. 

Reviewer #1

Line 478, what do you mean “as the pitch was kept constant”?

Authors: We appreciate for raising this point. Our intention it was to note that it would be expected to decrease the distance as more players were added to the same space. However, we decided to delete that specific part from the text to avoid misconceptions. 

Reviewer #1

Line 523, remove “available”.

Authors: Thank you. It was deleted accordingly. 

Reviewer #1

Line 500, revise “balls” to “ball”.

Authors: We appreciate for this suggestion. It was modified accordingly. 

Reviewer #1

Concise the sentences, line 513-516 and conclusion.

Authors: We attempted to be more concise in the lines 513-516, however, we are unsure if we were modified in the proper space. Sorry if we fail to correct the proper section. In addition, also the conclusion was also modified to become shorter. 

Reviewer #2

I would like to congratulate the authors with the final manuscript. I believe they addressed all my comments/suggestions and I think the final version is really good. Well done!

Dear reviewer, again, we would like to say thank for your contribution and expertise that helped us to improve the current work.

---

## [Decision Letter · Decision Letter 2]

3 Jun 2022

Amplifying the effects of adding extra players during association football game-based scenarios

PONE-D-22-04617R2

Dear Dr. Coutinho,

We’re pleased to inform you that your manuscript has been judged scientifically suitable for publication and will be formally accepted for publication once it meets all outstanding technical requirements.

Kind regards,

Rabiu Muazu Musa, PhD

Academic Editor

PLOS ONE

Additional Editor Comments (optional):

Reviewers' comments:

Reviewer's Responses to Questions

**Comments to the Author**

1. If the authors have adequately addressed your comments raised in a previous round of review and you feel that this manuscript is now acceptable for publication, you may indicate that here to bypass the “Comments to the Author” section, enter your conflict of interest statement in the “Confidential to Editor” section, and submit your "Accept" recommendation.

Reviewer #1: All comments have been addressed

2. Is the manuscript technically sound, and do the data support the conclusions?

Reviewer #1: Yes

3. Has the statistical analysis been performed appropriately and rigorously? 

Reviewer #1: Yes

4. Have the authors made all data underlying the findings in their manuscript fully available?

Reviewer #1: Yes

5. Is the manuscript presented in an intelligible fashion and written in standard English?

Reviewer #1: Yes

6. Review Comments to the Author

Reviewer #1: (No Response)

---

## [Editor Report · Acceptance letter]

14 Jun 2022

PONE-D-22-04617R2 

Amplifying the effects of adding extra players during association football game-based scenarios 

Dear Dr. Coutinho:

I'm pleased to inform you that your manuscript has been deemed suitable for publication in PLOS ONE. Congratulations! Your manuscript is now with our production department. 

Kind regards, 

on behalf of

Dr. Rabiu Muazu Musa 

Academic Editor

PLOS ONE